# Changes in the Epidemiology of Hepatobiliary Disease Complicating Type 2 Diabetes over 25 Years: The Fremantle Diabetes Study

**DOI:** 10.3390/jcm9113409

**Published:** 2020-10-24

**Authors:** Timothy M.E. Davis, Kirsten E. Peters, S. A. Paul Chubb, Leon A. Adams, Gary P. Jeffrey, Wendy A. Davis

**Affiliations:** 1University of Western Australia Medical School, Fremantle Hospital, Fremantle 6959, WA, Australia; kirsten@proteomics.com.au (K.E.P.); wendy.davis@uwa.edu.au (W.A.D.); 2Proteomics International, Nedlands 6009, WA, Australia; 3PathWest Laboratory Medicine Western Australia, Fiona Stanley Hospital, Murdoch 6150, WA, Australia; Paul.Chubb@health.wa.gov.au; 4University of Western Australia Medical School, Sir Charles Gairdner Hospital, Nedlands 6009, WA, Australia; leon.adams@uwa.edu.au (L.A.A.); gary.jeffrey@uwa.edu.au (G.P.J.); 5Department of Hepatology, Sir Charles Gairdner Hospital, Nedlands 6009, WA, Australia

**Keywords:** type 2 diabetes, hepatobiliary disease, non-alcoholic fatty liver disease, incidence

## Abstract

Objective: To determine whether the incidence/outcome of hepatobiliary disease (HBD) has increased over recent decades in community-based Australians with and without type 2 diabetes (T2D). Methods: Longitudinal data from the Fremantle Diabetes Study Phase I (FDS1; recruitment 1993–1996; *n* = 1291 with T2D) and Phase II (FDS2; 2008–2011; *n* = 1509) were analyzed. Participants with T2D from both Phases were age-, sex-, and postcode-matched 1:4 to people without diabetes. Incident HBD and associated mortality were ascertained from hospitalization, cancer registration, and/or death certification codes. Incidence rates (IRs) and IR ratios (IRRs) for those with versus without diabetes in FDS1 and FDS2 were calculated. Results: HBD IRs for people without diabetes did not change between Phases. The IRR (95% CI) for people with T2D in FDS2 versus FDS1 was 1.30 (1.01–1.68) with the highest IRRs in participants aged <65 years. Non-alcoholic fatty liver disease/steatohepatitis (NAFLD/NASH) events were 54% greater in FDS2 than FDS1 in the presence of greater abdominal adiposity. NAFLD/NASH was coded in one in 11 HBD events in FDS2 and in 10% of HBD deaths (<4% of total mortality). Conclusions: HBD is more frequent in people with versus without T2D and this discrepancy is increasing. Hospitalizations/deaths due to NAFLD/NASH remain uncommon.

## 1. Introduction

There have been significant changes in the epidemiology of complications of type 2 diabetes (T2D) over the past few decades. Studies from the US, Europe, and Asia have shown declines in cardiovascular disease (CVD) and microangiopathy reflecting, in large part, improved CVD risk factor management [1,2,3]. Recently published Australian data collected over 25 years from our representative community-based longitudinal Fremantle Diabetes Study Phase I (FDS1) and II (FDS2) show the same trends [4]. This means that people with T2D are now living long enough to develop nonvascular complications with major consequences. One of the most prominent among these is likely to be hepatobiliary disease (HBD), especially non-alcoholic fatty liver disease (NAFLD).

Diabetes is associated with an increased risk of a variety of HBDs. The prevalence of NAFLD in T2D is estimated to be approximately double that in the general population [5], and there have been concerns that a ‘tsunami’ of diabetes-related NAFLD complications including non-alcoholic steatohepatitis (NASH), cirrhosis, and hepatocellular carcinoma (HCC) is imminent in part because of the increasing incidence of T2D fueled by the obesity epidemic [6]. Gallbladder disease is also associated with diabetes independent of obesity in most relevant studies [7]. There have been reports of a high risk of death due to alcohol-related liver disease in men with diabetes [8], while viral hepatitis [9,10] and genetic hemochromatosis [11] have also been considered to increase the risk of liver disease complicating diabetes.

Specific evidence of a link between T2D and HBD comes from the FDS1 [12] and from a Canadian administrative database study conducted between 1994 and 2006 [13]. In the latter study, there was a hazard ratio of 1.77 for the combined endpoint of liver cirrhosis, liver failure, or liver transplantation in newly diagnosed people with diabetes of unspecified type. The findings in the FDS1 from data collected between 1993 and 2010 showed there was a significantly increased incidence rate ratio (IRR) of 1.66 for hospitalizations and deaths for/from a wider range of HBD in 1294 participants with T2D compared to 5156 matched individuals without diabetes followed for a mean of 11.5 years from recruitment in 1993–1996 [12]. However, only 7.6% of incident HBD events in the FDS1 participants with T2D were attributable to NAFLD.

Given the shift in complications away from vascular disease over recent decades [1,2,3,4] and the increasing obesity in developed countries such as Australia [14], we postulated that the incidence of HBD, and particularly NAFLD, would have increased since the FDS1 was conducted, in line with global predictions [6]. The aim of the present study was, therefore, to determine the incidence of hospitalizations and deaths for/from HBD in participants with T2D in the FDS2 cohort and in matched individuals without diabetes recruited 15 years after the equivalent groups in the FDS1.

## 2. Materials and Methods

### 2.1. Participants, Epidemiologic Setting, and Approvals

The FDS2 is an observational, longitudinal study of people with known diabetes conducted in a zip code-defined geographic area surrounding the port city of Fremantle in the state of Western Australia (WA) [15]. The FDS2 utilized the same basic design as FDS1 for identification and recruitment of participants with diabetes and matched residents from the same catchment area who did not have diabetes. During a three-year recruitment period between 2008 and 2011, individuals with diabetes (except gestational diabetes) were identified from hospital, clinic, and primary care patient lists; widespread advertising through local media, pharmacies, optometrists, and networks of health care professionals; and, in the case of FDS2 but not FDS1, third-party mail-outs to registrants of the Australian National Diabetes Services Scheme and the National Diabetes Register. Full details of recruitment, the final FDS2 sample characteristics including classification of diabetes types, and nonrecruited people identified with diabetes in the catchment area have been published [15]. The FDS2 protocol was approved by the Human Research Ethics Committee of the Southern Metropolitan Area Health Service (07/397 18 October 2007). All participants gave written, informed consent.

In FDS2, 4639 people with diabetes were identified from a population of approximately 157,000, and 1668 (36%) recruited of whom 1509 (90%) had T2D. Four age-, sex-, and postcode-matched residents (*n* = 6036) with no coding of diabetes before study entry on any WA health database were randomly selected from the study catchment area for each FDS2 participant at the time of their enrolment using the WA Electoral Roll and the WA Registry for Births, Deaths, and Marriages as a source of all residents. The present analyses excluded types of diabetes other than T2D. The follow-up of the matched residents was censored if and when they developed diabetes.

### 2.2. Clinical Assessment and Laboratory Tests

At baseline and three biennial face-to-face reviews up to 2017, a medical questionnaire was completed, a physical examination was performed, and fasting biochemical tests were carried out in a single, nationally accredited laboratory. Self-reported alcohol consumption was recorded, as were details of prior illnesses including HBD. Ethnic background was assessed as Anglo-Celt, Southern European, Other European, Asian, Aboriginal, or Other. Microvascular and macrovascular complications of diabetes were identified using standard criteria [16]. Face-to-face assessments were interspersed with three biennial postal questionnaires to supplement data collection.

### 2.3. Hospitalizations, Mortality, and Cancer Ascertainment

All hospitalizations, deaths, and cancer registrations in WA are recorded in the WA Data Linkage System (WADLS) [17], which was used to provide FDS2 participant outcomes from January 1980 until the end of December 2016. Incident HBD was taken as hospitalization/cancer registration for a diagnosis (principal or secondary) of (1) liver disease, (2) viral hepatitis, (3) malignant primary neoplasm of liver and intra-hepatic bile ducts, and (4) disorders of gall bladder and biliary tract as per relevant International Classification of Disease (ICD)-9 Clinical Modification (CM) and ICD-10 Australian Modification (AM) codes (see Table 1). NAFLD (ICD-10: K76.0) and NASH (ICD-10: K75.8) are not specifically defined within ICD-9 and so consideration of these conditions was limited to post-introduction of ICD-10 coding in Australia (on 1 July 1999). Causes of death were reviewed independently by two FDS physician investigators and classified under the system used in the UK Prospective Diabetes Study [18]. In the case of discrepant coding, case notes were consulted and a consensus obtained. Deaths attributable to NAFLD-associated cirrhosis were those in which (1) NAFLD or associated terminology was part of the reported causes of death, (2) another cause for cirrhosis such as alcohol or viral hepatitis was not reported, and/or (3) NAFLD was considered likely based on longitudinal FDS2 and other premorbid data. Secondary liver cancer and its sequelae and liver failure not attributable to HBD were excluded from the analyses.

### 2.4. Statistical Analysis

The computer packages IBM SPSS Statistics 25 (IBM Corporation, Armonk, NY, USA) and StataSE 15 (StataCorp LP, College Station, TX, USA) were used for statistical analysis. Data are presented as proportions, mean ± SD, geometric mean (SD range), or, in the case of variables that did not conform to a normal or log-normal distribution, median [interquartile range, IQR]. Two-sample comparisons were by Fisher’s exact test for proportions, by Student’s *t*-test for normally distributed variables, and Mann–Whitney U-test for non-normally distributed variables. Overall and 10-year age- and sex-specific incident rates for HBD were compared (1) in those with T2D in FDS1 and FDS2, and (2) for FDS2, in those with T2D and no diabetes, and respective IRRs derived. Overall IRRs for those aged >25 years were also estimated.

For the FDS2 type 2 diabetes cohort, Cox regression with age as the timeline and backward conditional variable entry (*p* < 0.05) and removal (*p* ≥ 0.05) was used to identify independent determinants of the first episode of HBD during follow-up from clinically plausible baseline variables at *p* < 0.20 in bivariable analyses. Log-normally distributed data were log (ln) transformed before analysis. The validity of the proportional hazards’ assumption was assessed by the examination of time-dependent covariates. Since some variables of interest were missing up to 20 values (1.5%), missing values were multiply imputed (×20), defining imputation models that included incident HBD. HBD component diagnoses were not included in similar Cox regression models because of their limited numbers.

## 3. Results

### 3.1. Baseline Participant Characteristics Including Prior Hepatobiliary Disease in Type 2 Diabetes

At study entry, the 1509 FDS2 participants with T2D had a mean age of 65.4 years, 51.8% were male and their median diabetes duration was 8.0 years. The characteristics of the FDS2 participants with T2D are compared with those for the equivalent FDS1 cohort in Table 2. The FDS2 participants were older, more likely to be male, had longer diabetes duration and were, thus, more likely to be insulin-treated, and they consumed more alcohol. Their cardiometabolic risk factors were better consistent with more intensive pharmacotherapy. Despite these differences, during the 28 to 31 years (average 29.5 years) leading up to study entry (from January 1980 to November 2008), 205 FDS2 participants (13.6%) had at least one hospitalization or cancer registration for/with HBD. This represented an average of seven patients or 0.5% of the cohort per year, figures similar to equivalent data from FDS1 participants with T2D at eight patients or 0.6% of the cohort per year [12].

### 3.2. Incident Hepatobiliary Disease in Type 2 Diabetes

During prospective follow-up of 8773 person-years (mean ± SD 6.7 ± 1.7 years) from entry to end-December 2016 or death (whichever came first), 150 FDS2 participants (9.9%) were hospitalized for HBD, registered with hepatobiliary cancer, or died from/with HBD, of whom 35 had a prior history. In order to capture new rather than potential repeat presentations of HBD diagnosed before recruitment, we assessed the 1304 participants without a history of HBD and found that 115 (8.8%) had their first episode during 8394 person-years (6.4 ± 2.0 (range 0.14–8.85) years) of follow-up (see Table 3), a crude incidence of 13.7/1000 patient-years. Using the equivalent incidence data from the FDS1 of 10.5/1000 person-years [12], the IRR (95% confidence interval (CI)) for FDS2 versus FDS1 was 1.30 (1.01, 1.68) (see Table 3).

There were no cases of HBD in FDS2 participants with T2D who were younger than 30 years. The incidence was highest in the 25–34-year age group (144.1 (95% CI 17.5, 520.4)/1000 person-years), followed by the 45–54 year (20.1 (11.3, 33.2)/1000 person-years) and ≥85 year (19.9 (11.8, 42.2)/1000 person-years) age groups (see Table 3). Compared with equivalent FDS1 data [12], the largest difference was in the 45–54-year age group in FDS2, which had an IRR of 2.77 (1.13, 7.17), an extra 12.9 (1.6, 24.1) cases per 1000 person-years (see Table 3).

The HBD-related variables for participants in FDS2 without a prior history of HBD are compared by incident HBD status in Table 4. Those with incident HBD were more likely to be of Aboriginal background, to be centrally obese, and to have Stage 4 or 5 renal disease than those without incident HBD. They also had lower platelet counts and higher serum gamma-glutamyl transferase concentrations at baseline.

In a Cox regression model with age as the timeline, independent determinants of new-onset incident HBD comprised Australian Aboriginal background, systolic blood pressure, body mass index, A Body Shape Index (ABSI) [19], severe renal impairment (stage 4 or 5 chronic kidney disease), ln(platelet count), and ln(gamma glutamyl transferase) (see Table 5). The proportional hazards assumption was not violated for any variable (*p* > 0.07).

### 3.3. Incident Non-Alcoholic Fatty Liver Disease-Related Hospitalization in Type 2 Diabetes

Before study entry but after the introduction of ICD-10 coding on 1 July 1999, 14 (0.9%) participants with T2D were hospitalized for/with NAFLD or NASH and were excluded from further analysis. Between study entry and the end of 2016, 13 FDS2 participants with T2D were hospitalized for the first time for/with NAFLD on 14 occasions but on only one occasion was NAFLD the principal diagnosis. Five participants were hospitalized for NASH as a secondary diagnosis after study entry, one of whom had been hospitalized three years earlier with NAFLD, and another two had a co-secondary diagnosis of NAFLD. Consistent with the approach adopted for all HBD, the number of FDS2 participants with a first presentation of NAFLD/NASH during 9921 person-years of follow-up was 16, or an incidence of 1.6 (0.9, 2.6)/1000 person-years. Excluding those with any prior HBD, 12 cases of NAFLD/NASH contributed to 10.4% of the overall HBD incidence in FDS2 participants with T2D. Although we reported that 7.6% of incident HBD events were attributable to NAFLD in the FDS1 participants with T2D [12], direct comparisons between the two study phases were difficult because of the lack of specific NAFLD coding in the ICD-9 system that was in use prior to 1999.

### 3.4. Incident Hepatobiliary Disease in Matched People without Diabetes

The 6036 matched residents without diabetes in FDS2 had a mean age of 65.4 ± 11.7 years and 51.8% were males. They were followed for a total of 40,013 person-years (mean ± SD 6.6 ± 1.8 (range 0.0–8.8) years) and 441 (7.3%) were hospitalized for HBD or registered with hepatobiliary cancer or died from/with HBD, of whom 270 had a prior history of HBD. In the 5595 participants with no prior history, 226 (4.0%) had their first episode of HBD during 36,600 person-years (6.5 ± 1.8 (range 0.0–8.8) years) of follow-up, a crude incidence of 6.2/1000 person-years (see Table 6). The FDS1 cohort without diabetes had a similar overall crude incidence rate at 6.4/1000 person-years [12], with an IRR of 0.97 (0.82, 1.14) (see Table 6).

There was one case of HBD in a participant without diabetes who was younger than 30 years. The incidence was highest in the 15–24-year age group (17.1 (0.4, 95.1)/1000 person-years), followed by the 25–34-year age group (15.4 (1.9, 55.8)/1000 person-years), and in those aged ≥85 years (13.2 (8.8, 19.1)/1000 person-years) (see Table 6). Compared with FDS1 data, the 45–54-year age group in the FDS2 participants without diabetes had a significantly higher incidence rate (2.42 (1.17, 5.24) or an extra 3.5 (0.7, 6.3) cases/1000 person-years) (see Table 6).

### 3.5. Incident Non-Alcoholic Fatty Liver Disease Hospitalizations in Matched People without Diabetes

Before study entry but after the introduction of ICD-10-AM coding on 1 July 1999, 13 (0.2%) matched residents without diabetes were hospitalized for/with NAFLD or NASH and were excluded from the following analyses. Between entry and end-2016, 16 matched residents without diabetes and no prior hospitalization for NAFLD or NASH were hospitalized for/with NAFLD or NASH on 29 occasions (19 with NASH including two as principal diagnosis) and 10 NAFLD (including one as principal diagnosis) during 39,927 person-years, an incidence of 0.4 (0.2, 0.9)/1000 person-years.

### 3.6. Incident Hepatobiliary Disease in People with Type 2 Diabetes Versus Those without Diabetes

In the comparison of FDS2 participants with T2D and those without diabetes, the IRR for new-onset HBD was 2.23 (1.76, 2.80), with the highest IRR in the 45–54-year age group (3.39 (1.64, 6.84)) (see Table 7). These data parallel those from FDS1 (see Table 8) [12], in which the IRR was 1.66 (1.36, 2.01), peaking at 2.97 (1.12, 7.52) for those aged 45–54 years.

For FDS2 participants without prior hospitalization for/with NAFLD/NASH after 1 July 1999, the IRR for new-onset NAFLD/NASH for those with T2D versus those without diabetes was 4.02 (1.88, 8.60) with an incident rate difference (IRD) of 1.21 (0.40, 2.03)/1000 person-years.

### 3.7. Deaths from Hepatobiliary Disease

By end-December 2016, 269 (17.8%) of the 1509 FDS2 participants with T2D had died, 10 (3.7% of deaths or 0.7% of the total cohort) with HBD as a cause of death or as an antecedent or a contributing factor (excluding secondary liver cancer and its sequelae, and liver failure not attributable to HBD), an incidence rate of 0.99 (0.47, 1.82)/1000 person-years. Of these, four (40.0%) died from/with primary hepatobiliary cancer, five (50.0%) from/with alcohol-related liver disease, and one (10.0%) from/with non-alcohol-related cirrhosis due to autoimmune hepatitis with NAFLD a contributing factor.

For those without diabetes in FDS2, 730 (12.1%) died during follow-up, 27 of HBD (3.7% of deaths or 0.4% of the total cohort), an incidence rate of 0.67 (0.44, 0.98)/1000 person-years. Of these, 10 (37.0%) died from/with primary hepatobiliary cancer, nine (33.3%) from/with alcohol-related liver disease, and the remaining eight (29.6%) from/with other causes (cholangitis, cholecystitis, decompensated liver failure due to chronic viral hepatitis C and alcohol abuse, other hepatitis). Twenty-five of the 27 deaths for/with HBD (92.6%) were documented as having liver disease in hospital records and/or the cancer register. The crude mortality ratio for death for/with HBD in the participants with T2D versus those without diabetes in FDS2 was 1.46 (0.63, 3.12). In the FDS1 the equivalent crude mortality ratio was 1.97 (1.16, 3.32) [12].

## 4. Discussion

The present detailed epidemiologic data show that the burden of HBD has increased over the last few decades in community-based Australian adults with T2D. Participants with T2D in the FDS2 were 30% more likely to be hospitalized for HBD, registered with hepatobiliary cancer, or to have died from/with HBD compared with those from the FDS1 recruited on average 15 years earlier. In addition, HBD was over twice as frequent during follow-up in FDS2 participants with T2D relative to a matched group of people without diabetes from the same geographic area, more than double the excess HBD risk in T2D versus no diabetes seen with the same comparison in FDS1 [12]. These findings, and the significantly greater BMI and waist circumference in FDS2 participants with T2D versus those in FDS1, support the notion that an increase in HBD complicating T2D is one of the legacies of the obesity epidemic, albeit that the average annual increase in HBD events over the 15 years between FDS1 and FDS2 was 2%/year. This is of concern but, notwithstanding that obesity-related liver disease is one part of HBD as a whole, it appears to fall short of the predicted ‘tsunami’ [6].

The age-specific IRRs indicated that the greatest disparity in HBD events for participants with T2D in FDS2 versus FDS1 was in those aged <65 years. This finding is in accord with US epidemiologic data showing that middle-aged adults have accounted for an increasing proportion of diabetes-related, largely vascular, complications over recent decades [20]. The same pattern was evident in the present study for the comparison between the two matched cohorts without diabetes in FDS2. This suggests that increasing obesity at younger age groups in the general population in Australia [14] and other developed countries such as the US [21] may have parallel implications for HBD rates for those without diabetes.

There were a number of independent baseline predictors of incident HBD events in FDS2 participants in a Cox proportional hazards model with age as the time scale. These included Aboriginal ethnicity, consistent with previous reports of an increased risk of hospitalization for cirrhosis [22] and of hepatobiliary cancer [23,24] in Australian indigenous communities. Both BMI and ABSI were in the model, confirming the strong link between obesity and HBD, with the ABSI as an indicator of more central concentration of body mass than BMI [19] having a comparatively stronger association. Although hypertension is a feature of the Metabolic Syndrome, which includes central adiposity and an increased risk of NAFLD [25], it has long been recognized that chronic liver disease can be associated with low blood pressure [26], consistent with the inverse relationship between systolic blood pressure and HBD events in the present study. The association with severe renal impairment can occur in the absence of significant alterations in renal histology (pre-renal), but intrinsic renal abnormalities can also complicate chronic liver disease [27]. A raised serum gamma glutamyl transferase and low platelet count and are found commonly in chronic liver disease, and their ratio has been suggested as a prognostic index for the development of fibrosis and cirrhosis [28].

Within the limitations of temporal changes in the ICD system that complicated direct comparisons between FDS phases, the percentage of HBD events attributable to NAFLD/NASH in FDS2 was 54% greater than in FDS1 and the IRR versus no diabetes in FDS2 was greater than 4. These observations are consistent with the increase in central adiposity between FDS phases and the relatively high prevalence of NAFLD in T2D compared with that in the general population [5]. However, NAFLD/NASH was coded in only one in 11 HBD events in FDS2 and represented a similar percentage (10%) of the low rate of deaths from HBD (<4% of total mortality). As with HBD events as a whole, the between-phase increase in NAFLD/NASH in T2D is concerning but appears to fall short of the predicted surge in diabetes-related NAFLD complications [6].

There are a number of possible reasons why our epidemiologic data do not align with predictions. First, the glitazones and newer therapies for T2D including the glucagon-like peptide 1 receptor agonists (GLP-1RA) and sodium–glucose co-transporter-2 inhibitors (SGLT2i) have evidence of benefit in NAFLD/NASH [29]. However, few of our participants (<1%) in FDS1 and FDS2 were taking a glitazone, and the use of GLP-1RA and SGLT2i use in FDS2 was similarly low given that they were marketed in Australia relatively late in the period of follow-up. Renin-angiotensin system (RAS) blocking drugs may have a role in attenuating the deleterious effects of RAS activation on the liver [30]. The use of these agents increased between FDS phases and may have limited an increase in NAFLD/NASH events over time. Lipid-modifying agents including statins and fibrates, which increased considerably between FDS phases, have also been suggested as influencing hepatic fat accumulation and fibrosis but the data are inconsistent [31]. There is evidence that improved glycemic control, as seen in FDS2 relative to FDS1, might itself reduce the progression of NAFLD/NASH [32]. An examination of the probably complex relationship between these factors and temporal changes in HBD, including NAFLD/NASH, was beyond the scope of the present study.

Current guidelines suggest that liver enzymes, steatosis biomarkers, and liver ultrasonography be considered for all people with T2D based on the need to identify individuals with NAFLD who are at risk of progression to NASH and its adverse consequences [33,34]. Given the present data, which suggest that these people form a relatively small subgroup of the total T2D population, as well as the high direct and indirect costs of such testing, the relatively low predictive value of non-invasive tests, the risks of liver biopsy, and the current paucity of effective treatments [35], future detailed analyses of longitudinal population-based studies of representative samples such as FDS2 could be useful in generating algorithms for simple cost-effective NALFD/NASH screening in usual care. Such algorithms could identify those people with T2D who are at risk of adverse outcomes and thus warrant further, more intensive investigation including specialized biochemical tests and imaging.

The present study had limitations. Although the use of administrative data to identify true cases of HBD has had variable reliability in previously published studies [36,37], the WADLS is regularly validated [17]. In addition, the known relationship between T2D and HBD including NAFLD/NASH may have meant that coding errors favored greater HBD ascertainment in individuals with diabetes. This would mean that the true difference between participants with T2D and without diabetes was less than we observed. We performed restricted multivariable analyses as there were limited data available for people without diabetes in both cohorts, and the number of outcomes in individual components of HBD including NAFLD/NASH was also limited. The strengths of the study include the representative, community-based participant samples and detailed participant-level data for those with T2D.

## 5. Conclusions

The present data from FDS2, complemented by previously published relevant FDS1 findings [12], show that there has been a relatively modest increase in the incidence of HBD events (including NAFLD/NASH) ascertained from administrative data in people with T2D compared to matched individuals without diabetes from the same geographical area over the past two decades. This change parallels evidence of increased visceral adiposity but does not suggest that the decline in the incidence of chronic cardiovascular complications of T2D is unmasking a substantial surge in HBD as obesity levels rise.

## Figures and Tables

**Table 1 jcm-09-03409-t001:** International Classification of Disease (ICD)-9 Clinical Modification (CM) and ICD-10 Australian Modification (AM) codes used to identify hepatobiliary disease outcomes in the Hospital Morbidity Data Collection.

Outcome	Diagnosis Codes		Procedure Codes	
	ICD-9-CM	ICD-10-AM	ICD-9-CM	ICD-10-AM
Liver cancer	155	C22		
Gall bladder cancer	156	C23, C24		
Hepatitis	070	B15-B19		
Liver disease	570–573	K70-K77		
Gall bladder disease	574–576	K80-K83, K87.0		
Liver transplant	V42.7	Z94.4	50.5, 50.51, 50.59	90317-00

**Table 2 jcm-09-03409-t002:** Baseline characteristics of participants with type 2 diabetes in the Fremantle Diabetes Study Phase I (FDS1) and FDS Phase II (FDS2).

Variable	FDS1	FDS2	*p*-Value
Number	1291	1509	
Age at study entry (years)	64.0 ± 11.2	65.4 ± 11.7	<0.001
Sex (% male)	48.7	51.8	0.005
Alcohol consumption (standard drinks/day)	0 [0–0.8]	0.1 [0–1.2]	<0.001
Diabetes duration (years)	4.0 [1.0–9.0]	9.0 [3.0–15.8]	<0.001
Body mass index (kg/m²)	29.6 ± 5.4	31.2 ± 6.1	<0.001
Central obesity (by waist circumference; %) ^*^	64.5	71.4	<0.001
Diabetes treatment (%):			<0.001
Lifestyle measures alone	31.9	24.1	
Oral agents/non-insulin injectables	55.7	53.4	
Insulin only	9.5	5.9	
Insulin + oral agents/non-insulin injectables	2.8	16.6	
Fasting serum glucose (mmol/L)	8.3 (5.9–11.5)	7.6 (5.6–10.2)	<0.001
HbA_1c_ (%)	7.3 (5.9–9.2)	7.1 (5.9–8.5)	<0.001
HbA_1c_ (mmol/mol)	56 (41–77)	54 (41–69)	<0.001
Systolic blood pressure (mmHg)	151 ± 24	146 ± 22	<0.001
Diastolic blood pressure (mmHg)	80 ± 11	80 ± 12	0.51
Antihypertensive medication (%)	50.9	73.7	<0.001
Total serum cholesterol (mmol/L)	5.4 (4.4–6.5)	4.2 (3.3–5.4)	<0.001
Serum HDL-cholesterol (mmol/L)	1.01 (0.75–1.38)	1.19 (0.92–1.55)	<0.001
Serum total:HDL-cholesterol ratio	5.3 (3.8–7.4)	3.5 (2.6–4.8)	<0.001
Serum triglycerides (mmol/L)	2.2 (1.2–3.9)	1.5 (0.9–2.6)	<0.001
Lipid-modifying medication (%)	10.5	68.5	<0.001

* ≥102 cm in men and ≥88 cm in women.

**Table 3 jcm-09-03409-t003:** Age-specific incidence rates (IRs) per 1000 person-years, incidence rate ratios (IRRs), and incident rate differences (IRDs) for first-ever hospitalization for/with or cancer registration or death from/with hepatobiliary disease (HBD) in type 2 diabetes for the Fremantle Diabetes Study Phase II (FDS2) versus the Fremantle Diabetes Study Phase I (FDS1).

Age Group (Years)	25–34	35–44	45–54	55–64	65–74	75–84	≥85	All
FDS2								
Number of HBD events	2	1	15	27	28	33	9	115
Years of follow-up	14	206	746	2022	2785	2156	453	8381
IR	144.1	4.9	20.1	13.4	10.1	15.3	19.9	13.7
FDS1								
Number of HBD events	0	2	9	22	55	46	10	144
Years of follow-up	50	386	1237	2960	4930	3421	685	13,668
IR	0	5.2	7.3	7.4	11.2	13.4	14.6	10.5
IRR	-	0.94	2.77	1.80	0.90	1.14	1.38	1.30
95% CI (exact)		0.02, 18.0	1.13, 7.17	0.99, 3.31	0.55, 1.45	0.71, 1.82	0.50, 3.78	1.01, 1.68
IRD/1000 person-years	144.1	−0.34	12.9	5.9	−1.1	1.9	5.5	3.2
95% CI (approximate)	−55.6, 343.7	−12.3, 11.6	1.6, 24.1	0, 11.8	−5.9, 3.7	−4.6, 8.4	−10.3, 21.2	0.17, 6.3

**Table 4 jcm-09-03409-t004:** Baseline characteristics of type 2 diabetes participants in the Fremantle Diabetes Study Phase II who had no prior history of hepatobiliary disease by incident hepatobiliary disease status. Data are percentages, mean ± SD, geometric mean (SD range) or median [interquartile range].

Variable	No Incident Hepatobiliary Disease	Incident Hepatobiliary Disease	*p*-Value
Number (%)	1189 (91.2)	115 (8.8)	
Age (years)	65.6 ± 11.5	64.8 ± 12.5	0.51
Sex (% male)	53.5	53.9	>0.99
*APOE4* allele (%)	24.1	19.5	0.42
Ethnic background (%):			0.006
Anglo-Celt	52.5	51.3	
Southern European	12.9	15.7	
Other European	8.3	3.5	
Asian	4.5	4.3	
Aboriginal	5.3	13.9	
Mixed/other	16.5	11.3	
Not fluent in English (%)	11.3	9.6	0.76
Currently married/*de facto* relationship (%)	64.3	56.5	0.11
Educated beyond primary level (%)	86.8	86.7	>0.99
Smoking status (%):			0.23
Never	43.7	38.6	
Ex	46.9	47.4	
Current	9.4	14.0	
Alcohol consumption (standard drinks/day)	0.1 [0–1.2]	0.1 [0–1.5]	0.65
Age at diabetes diagnosis (years)	55.9 ± 12.2	54.5 ± 14.1	0.32
Diabetes duration (years)	8.0 [2.3–15.1]	8.0 [2.2–16.0]	0.52
Diabetes treatment (%):			0.54
Diet	25.7	21.7	
Oral agents/non-insulin injectables	54.2	56.5	
Insulin only	4.5	7.0	
Insulin ± oral agents/non-insulin injectables	15.6	14.8	
Fasting serum glucose (mmol/L)	7.1 [6.1–8.7]	7.6 [6.5–9.6]	0.06
HbA_1c_ (%)	6.8 [6.2–7.7]	7.0 [6.2–7.9]	0.27
ABSI (m^11/6^ kg^−2/3^)	0.081 ± 0.005	0.083 ± 0.006	0.002
Body mass index (kg/m^2^)	31.0 ± 5.9	32.6 ± 7.3	0.027
Central obesity (by waist circumference; %)	69.5	80.7	0.013
Systolic blood pressure (mmHg)	146 ± 22	142 ± 21	0.053
Diastolic blood pressure (mmHg)	80 ± 12	80 ± 14	0.73
Heart rate (bpm)	69 ± 12	73 ± 14	0.004
Taking antihypertensive medication (%)	74.2	71.1	0.50
Total serum cholesterol (mmol/L)	4.3 ± 1.1	4.4 ± 1.2	0.31
Serum HDL-cholesterol (mmol/L)	1.24 ± 0.34	1.20 ± 0.34	0.19
Serum triglycerides (mmol/L)	1.5 (0.9–2.5)	1.7 (0.9–3.0)	0.030
Taking lipid-lowering medication (%)	69.4	66.7	0.60
On aspirin (%)	36.9	36.8	>0.99
Cerebrovascular disease (%)	8.0	9.6	0.59
Coronary heart disease (%)	26.3	30.4	0.38
Peripheral arterial disease (%)	21.2	28.1	0.10
Peripheral sensory neuropathy (%)	58.1	62.8	0.37
eGFR category^*^ (%):			0.011
≥90 mL/min/1.73m^2^	38.8	39.5	
60–89 mL/min/1.73m^2^	45.8	36.8	
45–59 mL/min/1.73m^2^	8.6	8.8	
30–44 mL/min/1.73m^2^	4.5	7.0	
<30 mL/min/1.73m^2^	2.3	7.9	
Urinary albumin:creatinine ratio (mg/mmol)	3.1 (0.9–11.4)	3.9 (0.8–20.4)	0.08
Any retinopathy (%)	36.0	36.0	>0.99
Anemia (%)	10.9	16.5	0.09
AST/ALT ratio	1.1 (0.8–1.6)	1.2 (0.9–1.7)	0.18
Platelets (×10^9^/L)	250 (193–323)	229 (164–321)	0.010
Serum albumin (g/L)	44 ± 3	44 ± 3	0.10
Serum gamma-glutamyl transferase (U/L)	30 (15–58)	41 (18–93)	<0.001
Serum bilirubin (µmol/L)	9.5 (6.3–14.4)	9.8 (6.2–15.5)	0.43
Serum apolipoprotein A-1 (g/L)	1.48 ± 0.26	1.47 ± 0.27	0.58
Serum hyaluronic acid (µg/L)	52 (25–106)	63 (30–135)	0.005
Serum haptoglobulin (g/L)	1.58 ± 0.57	1.55 ± 0.65	0.64
Serum alpha-2 macroglobulin (g/L)	2.04 (1.45–2.87)	2.25 (1.57–3.22)	0.004

* Estimated glomerular filtration rate (eGFR) based on the CKD-EPI (Chronic Kidney Disease Epidemiology Collaboration) equation.

**Table 5 jcm-09-03409-t005:** Cox regression model of independent associates of new-onset incident hepatobiliary disease in participants with type 2 diabetes in the Fremantle Diabetes Study Phase II who had no prior history of hepatobiliary disease, using age as the time scale.

Variable	Hazard Ratio (95% CI)	*p*-Value
Aboriginal ethnic background	2.67 (1.41, 5.06)	0.003
BMI (increase of 1 kg/m^2^)	1.03 (1.01, 1.06)	0.017
ABSI ^*^ (increase of 0.001 m^11/6^ kg^−2/3^)	1.06 (1.02, 1.10)	0.002
Systolic blood pressure (increase of 1 mm Hg)	0.99 (0.98, 0.999)	0.026
eGFR <30 mL/min.1.73m^2^	3.95 (1.88, 8.33)	<0.001
Ln(Platelets (×10^9^/L))	0.24 (0.12, 0.46)	<0.001
Ln(gamma-glutamyl transferase (U/L))	1.58 (1.23, 2.02)	<0.001

*A Body Shape Index (ABSI) [19].

**Table 6 jcm-09-03409-t006:** Age-specific incidence rates (IRs) per 1000 person-years, incidence rate ratios (IRRs), and incident rate differences (IRDs) for first-ever hospitalization for/with or cancer registration or death from/with hepatobiliary disease (HBD) in people without diabetes for the Fremantle Diabetes Study Phase II (FDS2) versus the Fremantle Diabetes Study Phase I (FDS1).

Age Group (Years)	15–24	25–34	35–44	45–54	55–64	65–74	75–84	≥85	All
FDS2									
Number of HBD events	1	2	5	22	42	65	75	28	225
Years of follow-up	59	130	937	3709	8970	11,783	8901	2117	36,547
IR	17.1	15.4	5.3	5.9	4.7	5.5	6.9	13.2	6.2
FDS1									
Number of HBD events	0	2	4	13	60	136	147	44	406
Years of follow-up	13	235	1637	5313	13,052	22,169	17,217	4302	63,925
IR	0	8.5	2.4	2.5	4.6	6.1	8.5	10.2	6.4
IRR	-	1.81	2.19	2.42	1.02	0.90	0.99	1.29	0.97
95% CI (exact)		0.13, 25.0	0.47, 11.0	1.17, 5.24	0.67, 1.54	0.66, 1.22	0.74, 1.31	0.78, 2.12	0.82, 1.14
IRD/1000 person-years	17.1	6.9	2.9	3.5	0.1	−0.6	−0.1	3.0	−0.2
95% CI (approximate)	−16.4, 50.5	−17.5, 31.4	−2.4, 8.2	0.7, 6.3	−1.8, 1.9	−2.3, 1.1	−2.5, 2.2	−2.8, 8.8	−1.2, 0.8

**Table 7 jcm-09-03409-t007:** Age-specific incidence rates (IRs) per 1000 person-years, incidence rate ratios (IRRs), and incident rate differences (IRDs) for first-ever hospitalization for/with or cancer registration or death from/with hepatobiliary disease (HBD) in the Fremantle Diabetes Study Phase II cohort for type 2 diabetes versus no diabetes.

Age Group (Years)	15–24	25–34	35–44	45–54	55–64	65–74	75–84	≥85	All
Type 2 diabetes									
Number of HBD events	0	2	1	15	27	28	33	9	115
Years of follow-up	15	14	206	746	2022	2785	2156	453	8381
IR	0	144.1	4.9	20.1	13.4	10.1	15.3	19.9	13.7
No diabetes									
Number of HBD events	1	2	5	22	42	65	75	28	225
Years of follow-up	59	130	937	3709	8970	11,783	8901	2117	36,547
IR	17.1	15.4	5.3	5.9	4.7	5.5	6.9	13.2	6.2
IRR	0	9.32	0.91	3.39	2.85	1.82	2.23	1.50	2.23
95% CI (exact)	-	0.68, 128.6	0.02, 8.12	1.64, 6.84	1.69, 4.74	1.13, 8.50	1.42, 3.47	0.6, 3.3	1.76, 2.80
IRD/1000 person-years	−17.1	128.5	−0.5	14.2	8.7	4.5	8.5	6.6	7.6
95% CI (approximate)	−50.5, 16.4	−72.1, 329.0	−11.1, 10.1	3.7, 24.7	3.4, 13.9	0.6, 8.5	3.0, 14.0	−7.2, 20.5	4.9, 10.2

**Table 8 jcm-09-03409-t008:** Age-specific incidence rates (IRs) per 1000 person-years, incidence rate ratios (IRRs), and incident rate differences (IRDs) for first-ever hospitalization for/with or cancer registration or death from/with hepatobiliary disease (HBD) in the Fremantle Diabetes Study Phase I cohort for type 2 diabetes versus no diabetes.

Age Group (Years)	25–34	35–44	45–54	55–64	65–74	75–84	≥85	All
Type 2 diabetes								
Number of HBD events	0	2	9	22	55	46	10	144
Years of follow-up	50	386	1237	2960	4930	3421	685	13,668
IR	0	5.2	7.3	7.4	11.2	13.4	14.6	10.5
No diabetes								
Number of HBD events	2	4	13	60	136	147	44	406
Years of follow-up	235	1637	5313	13,052	22,169	17,217	4302	63,925
IR	8.5	2.4	2.4	4.6	6.1	8.5	10.2	6.4
IRR	0	2.12	2.97	1.62	1.82	1.57	1.43	1.66
95% CI (exact)	-	0.19, 14.82	1.12, 7.52	0.94, 2.68	1.30, 2.51	1.11, 2.21	0.64, 2.88	1.36, 2.01
IRD/1000 person-years	−8.5	2.7	4.8	2.8	5.0	4.9	4.4	4.2
95% CI (approximate)	−20.3, 3.2	−4.8, 10.3	−0.1, 9.8	−0.5, 6.2	1.9, 8.2	0.8, 9.0	−5.2, 13.9	2.4, 6.0

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
