# Peer review of "Changes in the Epidemiology of Hepatobiliary Disease Complicating Type 2 Diabetes over 25 Years: The Fremantle Diabetes Study"

_jcm, 2020, doi:10.3390/jcm9113409_

Round 1

Reviewer 1 Report

In this manuscript, the authors studied whether the incidence/outcome of HBD has increased over recent decades in community-based Australians with and without T2D.  The authors analyzed longitudinal data from FDS1 and FDS2.  Their results suggest that HBD is more frequent in people with T2D, and the discrepancy is increasing.

This is an interesting study, in which the authors looked into the correlation between HBD and metabolic symptom.  The data size and the analysis seems convincing.  The manuscript is also very well prepared.

As the authors showed in their discussion, the study does have limitations including the known association between HBD and metabolic syndrome. However, the data which were analyzed in this study provide enriched information for the changes in the epidemiology of HBD in 25 years. The paper is well written and easy to read. The conclusions are consistent with the presented evidence, meanwhile, the authors are also aware of the limitations of their study. This manuscript should be interesting to researchers who are working on the relationship between HBD and metabolic dysfunction, and can be documented.

Author Response

Response: None required.

Reviewer 2 Report

Methods:

  • Line 72: have you excluded Type 1 diabetes patients? it is known that have a different risk of HBD complications.
  • Line 85: i think that the study would be more informative excluding those healthy controls who later developed diabetes since they may have hidden insulino-resistance and other risk factors which may bias you analysis, and the simple censoring of the time after diabetes start is not enough.
  • Line 99: I'm not sure that you may include viral hepatitis in HBD events

Results:

  • line 140: different analysis for each HBD are welcomed and more informative. Please correct.
  • PAge 6 and 7 please correct line numeration: In new line 30 please report incidence rates and IRR calculated with healthy people not experiencing subsequent diabetes.
  • Please perform in diabetes cohort multivariable analysis evaluating factors associated to HBD occurrence and NALFD related death and hospitalization.

Discussion

- It is not clear what are the new findingins of the present study and the possible clinical applications. Do you propose a screening for HBD in such patients? Are there useful and accurate screening methods? At this purpose you should mention which are the possible methods to assess HBD in T2D patients (doi: 10.1016/j.jhep.2015.11.004.; doi: 10.1586/17474124.2015.1049155.)

- Limitations of the study should be further address

- Please correct typos throughout the manuscript

Author Response

Methods:

  • Line 72: have you excluded Type 1 diabetes patients? it is known that have a different risk of HBD complications.

Response: Other types of diabetes including type 1 were excluded. This is now explicitly stated in the text but should be clear from the fact that T2D made up 90% of the cases in FDS2.

  • Line 85: i think that the study would be more informative excluding those healthy controls who later developed diabetes since they may have hidden insulino-resistance and other risk factors which may bias you analysis, and the simple censoring of the time after diabetes start is not enough.

Response: We would prefer to retain our policy of only excluding those controls (who are not necessarily ‘healthy’) without diabetes once they had been diagnosed. Apart from the fact that the number of these participants was small, it is possible that they developed non-T2D without antecedent insulin resistance (IR). In addition, immortal time bias would be introduced if we excluded those who first had diabetes coded after baseline from the matched cohort. There will be a significant proportion of the general population with IR and pre-diabetes, and an examination of the association between glucose intolerance short of diabetes and HBD was beyond the scope of the present study.

  • Line 99: I'm not sure that you may include viral hepatitis in HBD events

Results: We included viral hepatitis in HBD events, as in our first paper on FDS1 (reference #12) and other similar epidemiologic studies (e.g. reference #13). Most of these cases were hepatitis B or C which can clearly have short- and long-term consequences for liver pathology.

  • line 140: different analysis for each HBD are welcomed and more informative. Please correct.

Response: The aim of the present study was to determine whether, in view of the emerging obesity epidemic, HBD rates and, in particular, those of NAFLD/NASH, had changed in the 15 years between FDS phases. We are concerned that further subdividing HBD causes will result in inadequate statistical power because of small sub-group numbers of events and deaths (as seen in FDS1 - reference #12 - and already addressed in the limitations paragraph in the Discussion). Our view is that we should keep the content of the present paper consistent with the stated aim.   

  • PAge 6 and 7 please correct line numeration: In new line 30 please report incidence rates and IRR calculated with healthy people not experiencing subsequent diabetes.

Response: We would prefer, as outlined in our response to the Reviewer’s second comment above, to retain our policy of only excluding those controls without diabetes once they had been diagnosed. These rates would potentially suffer from immortal time bias, as described in the response above, and we would prefer not to include such a sensitivity analysis. There are issues with line numbering in the reformatted document which we were unable to rectify – apologies.

  • Please perform in diabetes cohort multivariable analysis evaluating factors associated to HBD occurrence and NALFD related death and hospitalization.

Response: We have, in response to this comment, included a new multivariable analysis of predictors of HBD events in the FDS2 participants with T2D. This is in Table 5 and is accompanied by additional Methods, Results and Discussion. There are two few NAFLD-related events for a similar analysis in this sub-group.

Discussion

- It is not clear what are the new findingins of the present study and the possible clinical applications. Do you propose a screening for HBD in such patients? Are there useful and accurate screening methods? At this purpose you should mention which are the possible methods to assess HBD in T2D patients (doi: 10.1016/j.jhep.2015.11.004.; doi: 10.1586/17474124.2015.1049155.)

Response: The new findings are that the ‘tsumani’ of HBD, and of NALFD/NASH in particular, have not materialized in a community setting in a developed country. Although we restricted our Discussion to the epidemiologic data and their interpretation, we have, in line with the Reviewer’s comment, included a new paragraph relating to the implications for screening in the Discussion which includes the two references the Reviewer has specified.

- Limitations of the study should be further address

Response: We consider that we have identified the main limitations of the study but would be happy to include others if the Reviewer can specify them.

- Please correct typos throughout the manuscript

Response: Typos have been corrected – we used English rather than US spelling which may be what the Reviewer is referring